# Characterisation of Dry-Salted Violino and Bresaola from *Grass-Fed* Bergamasca Sheep

**DOI:** 10.3390/ani14030488

**Published:** 2024-02-01

**Authors:** Annalaura Lopez, Edda Mainardi, Ernesto Beretta, Sabrina Ratti, Federica Bellagamba, Carlo Corino, Vittorio Maria Moretti, Raffaella Rossi

**Affiliations:** 1Department of Veterinary Medicine and Animal Science, Università Degli Studi di Milano, Via Dell’Università 6, 26900 Lodi, Italy; annalaura.lopez@unimi.it (A.L.); edda.mainardi@unimi.it (E.M.); ernesto.beretta@unimi.it (E.B.); federica.bellagamba@unimi.it (F.B.); carlo.corino@unimi.it (C.C.); vittorio.moretti@unimi.it (V.M.M.); 2Independent Researcher, 20110 Milan, Italy; sabrinaratti@hotmail.com

**Keywords:** sustainable farming system, *grass-fed*, sheep dry salted products, quality parameters

## Abstract

**Simple Summary:**

In recent years, consumers have paid more attention to animal welfare and sustainable farming, shifting their focus to products that come from a traceable and sustainable supply chain. This trend represents a potential expansion for the sheep meat market, as the small ruminant farming system has the capacity to have a positive impact on environmental sustainability. In Italy, the sheep sector plays a minor role in the national agricultural economy but is fundamental to the protection and conservation of marginal and mountain areas. Mountain grazing is the most common method of rearing Bergamasca sheep in northern Italy, and in this system the animals are fed exclusively on grass, with no additional feed supplements. In the sheep market, the demand for meat is mostly focused on young animals and a proper characterisation and valorisation of dry cured products from *grass-fed* Bergamasca sheep of high weight and age could help to increase their relevance in the market.

**Abstract:**

This study focuses on characterising two seasoned products, violino and bresaola, derived from *grass-fed* Bergamasca sheep monitored with a GPS system. The evaluation includes both nutritional and sensory aspects. Results reveal that both products boast a high protein content (approximately 40%) and a beneficial fatty acid profile, endorsing a healthy n-6/n-3 ratio (2.2), along with rumenic acid (92–184 mg/100 g) and branched-chain fatty acids (BCFAs) (237–621 mg/100 g). The sensory evaluation highlights distinctive ovine characteristics in aroma and flavour. Violino and bresaola from *grass-fed* Bergamasca sheep showcase positive attributes for human consumption. The study emphasises the potential for added value to seasoned products from Bergamasca sheep through a traceable, sustainable, and health-conscious supply chain.

## 1. Introduction

In recent years, there has been a remarkable shift among consumers towards an increased health consciousness and increased awareness of sustainable farming systems, driven in part by the widespread dissemination of information [1]. As a result, they are looking for products from sustainable supply chains that ensure both healthy products and animal welfare. This trend represents a potential expansion for the sheep meat market, as the small ruminant farming system has the capacity to positively impact several Sustainable Development Goals outlined by the United Nations [2].

In 2022, the EU production of sheep and goat meat was estimated at 0.5 million tonnes, 1.1% more than in 2021 [3]. Sheep meat represents about 90% of the total meat production, with a population of 50,000 thousand animals [4]. Sheep meat is produced in Spain (27.8%), France (18.2%), Ireland (14.5%), and Greece (14.3%). It is particularly important in some rural areas of Spain, Italy, France, and Greece in supporting the local economy [5]. In Italy, the current sheep population is approximately 6 million heads and includes several breeds with different production aptitudes. In the meat market, sheep and goats together contribute approximately 30,000 tons per year (94% sheep and 6% goats). Of this total, about 54% is composed of milking lambs, 29% adult or ewes at the end of their productive life, and 11% castrated and heavy lambs [6]. In Italy, the sheep and goat sectors play a minor role in the national agricultural economy, accounting for 1% of the milk production and 0.3% of the meat production. However, despite these lower percentages compared to other livestock production, the sector plays a fundamental role in the protection and maintenance of marginal and mountain areas [5]. In this market, the demand for sheep meat is mainly focused on young animals like lambs due to their sensory characteristics, and this consumption is concentrated during the Christmas and Easter periods, leading to fluctuations in profits [7]. Further difficulties come from the lack of innovation of the sector, which affects its ability to successfully expand emerging markets [6]. In addition, meat and derived products from sheep have a unique aroma and flavour that are usually only appreciated by consumers already familiar with them, but a correct characterisation and valorisation of these products could help to increase their relevance in the market [8]. 

The Bergamasca sheep is an autochthonous Italian breed classified as an alpine breed. It is characterised by large animals and is primarily bred for meat production [9]. Usually, the main products derived from the sheep breeding are lambs, which are slaughtered for fresh meat production at around six months of age at approximately 45 kg in weight. Traditionally, the finest hind cuts of older sheep slaughtered at 80–90 kg are used for various types of cured, salted, and seasoned products, such as violino and bresaola [10].

The transhumant system on alpine pasture has been the main method of rearing Bergamasca sheep because of the animal’s remarkable resilience and adaptability to different environments [11]. In fact, the transhumant system is highly environmentally sustainable due to the movement of the animals, which prevents overgrazing and allows time for the regeneration of the natural plant species. This approach facilitates the conservation of rich natural biodiversity in marginal and mountain areas [5]. In addition, in this system the animals are exclusively “*grass-fed*”, feeding on only grass and hay, with no additional feed supplement. Allowing the animals to graze on a daily basis throughout the year promotes their well-being and allows them to express their natural behaviour. This type of feeding is also associated with a higher quality of meat and derived products [12].

The nutritional and sensory qualities of ovine meat are influenced by several factors, including age of the animal, diet composition, sex, and breed [13,14,15]. In this context, processed sheep products, such as sausages or hams, have a high protein and lipid content and a fatty acid profile with a favourable polyunsaturated fatty acids (PUFAs) to saturated fatty acids (SFAs) ratio and n-6/n-3 ratio and low cholesterol levels, making them nutritionally balanced [16]. 

These products are dry-cured by artisan methods, with expert butchers playing a key role in the process. This meticulous and traditional method involves treating the meat surface with a mixture of salt, sugar, spices, and typically nitrates. This process usually takes a long time (3 months to a year) and results in a great amount of shrinkage, up to 45%, and weight loss of the product from drying out the surface. Examples of dry-cured products are hams and other processed meat from different species [16,17].

This study aims to characterise dry-salted violino and bresaola derived from *grass-fed* Bergamasca sheep. By exploring the distinctive attributes of products from this autochthonous Italian breed, particularly those derived from older ewes, this research seeks to contribute to the understanding and appreciation of traditional sheep products. Additionally, the investigation intends to shed light on the sustainable and environmentally friendly practices associated with the transhumant system on alpine pastures, emphasising the impact on product quality and nutritional properties. The findings aim to provide valuable insights for both consumers seeking sustainable choices and producers aiming to enhance the market presence of traditional sheep meat products.

## 2. Materials and Methods

### 2.1. Study Area

The province of Bergamo is situated in the central-eastern part of Lombardy, Italy, covering an area of 2745 km^2^. The terrain varies from 82 to 3050 m above sea level. The northern section of the province features a mountainous landscape, constituting 64% of the surface area and encompassing the primary Bergamo valleys: Seriana Valley, Brembana Valley, Imagna Valley, Scalve Valley, San Martino Valley, and Cavallina Valley.

Scalve Valley, where this study was conducted, spans approximately 20 km and is surrounded by a ring of mountains, including the Presolana Massif (2521 m), Pizzo Tornello (2687 m), Cimon della Bagozza (2409 m), and Pizzo Camino (2492 m). The elevations in Scalve Valley range from 980 m above sea level in the locality of Azzone to a maximum of 1140 m above sea level in Schilpario.

### 2.2. Animal Management

A flock consisting of around 1500 sheep of the Bergamasca breed was considered in this study. The duration of the grazing period was 69 days, from 24 June 2021 to 1 September 2021. The flock, composed of 1300 sheep and 200 lambs, was monitored by using a collar equipped with a GPS tracking system, with SAT communication. The nylon OVItrace collar (160 × 5 cm) was secured around the neck of the flock lead sheep and another sheep as a backup. This system consists of an integrated electronic card with GPS, SAT communications, microprocessor with memory, and an antenna packaged in an IP67 case. The case has an on/off switch, two status lights, and a micro-USB connector. The base version electronics is powered by 4 lithium I92 AAA batteries each with 1200 mAh and a nominal voltage of 1.5 V. The GPS position fix rate was set to once an hour to ensure a longer battery life, and this gives a battery life of 4 to 6 weeks in an average environment. Additional batteries in a larger case have extended the system life to around 6 months without a battery change. The data collection and presentation were managed by the standard platform offered by the hardware and connectivity supplier SPOT, LLC (Globalstar Inc., Covington, LA, USA) and the Findmespot.com website. 

### 2.3. Data Collection and Meat Product Sampling

At the end of the transhumance period, in September, ten sheep with a minimum age of 4 years and a live weight between 80 and 100 kg were slaughtered following the European regulations regarding animal welfare at slaughter [18]. Hot carcass weights were recorded, and the carcasses were chilled at 4 °C for 24 h. Following this, the carcasses were split into two halves from the pelvis to the neck along the vertebral column and the primary cuts were weighed. Subsequently, the left leg and the topside of the sheep (*n* = 10) were sent to the processing plant for the preparation of deboned violino and bresaola. The raw materials were weighed prior to processing and upon completion of the ripening process, the individual weights of the products were measured to calculate the weight loss.

### 2.4. Product Processing

Bresaola was obtained from the dry-salting of the topside cut. The salting was carried out at refrigeration temperatures in steel tanks, where the meat was mixed with a mixture of sodium chloride (20 g/kg meat), spices and aromatic plants, rosemary (*Salvia rosmarinus*), bay leaf (*Laurus nobilis*), juniper (*Juniperus L communis*), garlic (*Allium sativum*), and pepper (*Piper nigrum* L.). The combination of the juice released from the meat pieces and the mixture formed the brine. Salting lasted for 15 days at 60–70 RH, depending on the size of the pieces, and the meat underwent daily stirring operations within the tank to facilitate the migration of salt and flavours into the muscle. During this phase, the muscle lost part of its water content. After salting, the meat was dried for 10 days at 16 °C and under conditions of RH ranging from 60% to 40%.

The phase of ripening was carried out in rooms where air exchange, the right level of RH (60–70%), and an average temperature between 8 and 12 °C were ensured. Maturation, including the drying time, lasted 60 days. This period allows for further water loss and concentration of flavours that will then contribute to the typical taste and aroma of the product. 

The preparation of violino was quite similar, with extended processing times due to the greater size of meat cuts. Left sheep’s leg muscles (top sirloin, top round, outside round) were manually deboned and put into a steel tank with a salting mixture that, for 1 kg fresh meat, consisted of 20 g sodium chloride and 2.2 g aromatic herbs and spices (rosemary, bay leaf, juniper, garlic, and pepper). The muscles were left to rest in the container at 4–5 °C for 10 days with RH at 70%. When the salting period was over, the legs underwent a drying period of 15 days at 16 °C and 45% RH. Then, violino was brushed to remove the salt residues and ripened at a temperature of 12 °C and 75% RH for a period of 90 days.

### 2.5. Sampling of Seasoned Products and Physical Measurements

The violino and bresaola samples were weighed and sectioned perpendicular to the fibers’ length. Then, they were cut into slices (1 cm thick), vacuum-packaged and frozen at −20 °C until laboratory analyses. Half of the violino and half of the bresaola were vacuum-packaged and stored at 4 °C until sensory evaluation.

At sampling, pH and colour parameters were measured. The pH test was performed using a portable pH meter equipped with a meat-penetrating probe (HI98191 microcomputer; Hanna Instruments, Vila do Conde, Portugal) and calibrated with a standard buffer of pH 4.0 and 7.0. The pH value of the different products was measured by inserting the probe in duplicate.

The colour indexes, lightness (L*), redness (a*), and yellowness (b*), were measured using a CR-300 Chroma Meter (Minolta Camera, Co., Osaka, Japan). The instrument was calibrated using a white calibration plate (Calibration Plate CR-A43; Minolta Camera, Co., Osaka, Japan). The colorimeter had an 8 mm measuring area and was illuminated with a pulsed Xenon arc lamp (illuminat C) at a viewing angle of 0°. Reflectance measurements were obtained at a viewing angle of 0° and the spectral component was included. Each value is the mean of six replications at the product sample surface.

### 2.6. Proximate Composition

Prior to analysis, samples were thawed for 24 h at 4 °C. Fifty-gram portions of violino and bresaola samples were ground using a blade homogeniser. The chemical composition was determined according to the official methods of the Association of Official Analytical Chemists [19]. Determinations of moisture (method 985.41), ash (method 920.153), fat (method 960.39), and crude protein (method 928.08) content were performed in duplicate.

### 2.7. Fatty Acid Profile

The intramuscular lipids were cold-extracted using a chloroform–methanol solution according to the method of Folch et al. [20]. The fatty acid composition was determined by gas chromatography preceded by the preparation of the fatty acid methyl esters using a combination of sodium methoxide in methanol (1M) and acetyl chloride in methanol (1:10, *v*/*v*), according to the method of Perez-Palacios et al. [21].

The fatty acid methyl esters were separated using a gas chromatograph (TRACE 1300 GC with AI 1310 autosampler, Thermo Fisher Scientific, Waltham, MA, USA) equipped with a TRACE™ TR-FAME capillary column (Thermo Fisher Scientific; 60 m, internal diameter 0.25 mm, 0.25 µm stationary phase thickness), using helium as carrier at a flow rate of 1 mL/min. The injector temperature was set at 260 °C and the detector temperature (FID) at 280 °C. The oven temperature program was set as follows: initial temperature of 40 °C for 3 min followed by a temperature ramp of 2.5 °C/min until reaching 180 °C and rising to 210 °C at a rate of 2 °C/min; at 210 °C it remained stationary for 25 min. The total time of each run (one sample) was thus 99 min.

The identification of fatty acid methyl esters was carried out by comparing retention times with those of different mixtures of analytical standards: a mixture of 37 fatty acid methyl esters (Supelco FAME37, Supelco 37 component mix, Supelco, Sigma Aldrich, St. Louis, MI, USA), plus individual branched-chain fatty acid standards (anteiso13, iso13, iso14, anteiso14, iso15, anteiso15, iso16, anteiso16, iso17, anteiso17, iso18, anteiso18), oleic acid isomer standards (cis11-18: 1, trans11-18:1, trans9-18:1), and linoleic acid isomer standards (t10c12-18:2, c9t11-18:2, t9t12-18:2, c9t12-18:2, t9c12-18:2, c9c12-18:2) purchased by Larodan (Solna, Sweden). The methyl ester of C19:1 was used as an internal standard. The quantification of fatty acids was obtained by integration of the chromatographic peak areas and internal-standard-based calculations, according to the method described in Vahmani et al. [22]. Fatty acids were quantified in mg of fatty acid in 100 g of the product. To facilitate a meaningful comparison of fatty acid data, and to mitigate the impact of variation in lipid content between the two products, the results were additionally presented as a percentage of the total fatty acids.

### 2.8. Sensory Analysis

A selected and trained sensory panel, familiar with sheep meat and descriptive analysis procedures [23], of 10 judges (6 females and 4 males, with an age range from 25 to 50) was chosen. Three sessions were conducted to develop a common vocabulary and improve the ability of judges to discriminate between samples, as well as the correct use of the intensity scale. 

The final list of descriptors used for the violino consists of 16 attributes: 3 of appearance (red colour, white colour, and homogeneity of the slice), 4 of aroma (global intensity of aroma, pepper, seasoned, wild), 2 of taste (salty, sweet), 5 of flavour (global intensity of flavour, spicy, wild, pepper, seasoned), and 2 of texture in mouth (hard and soft). 

For bresaola, the profile consists of 17 attributes: 3 of appearance (red colour, white colour, and red colour homogeneity), 5 of aroma (global intensity of aroma, pepper, spicy, seasoned, wild), 2 of flavour (salty, sweet), 5 of flavour (global intensity of flavour, spicy, wild, pepper, seasoned), and 2 of texture in mouth (tender and soft fat).

All assessments were carried out in a sensory laboratory equipped according to ISO recommendations [24]. The judges evaluated violino and bresaola samples in triplicate, measuring the intensity of each attribute by scoring on a scale from 0 (no sensation) to 10 (extreme intensity) for the two different products. 

The judges were presented 1.5 mm thick slices of violino or bresaola provided on disposable plastic plates. They were instructed first to score the external appearance and aroma, then to take a slice and score the texture. During sampling, panel members had access to unlimited water and unsalted crackers.

### 2.9. Statistical Analysis

The sensory data for each attribute were submitted to analysis of variance (ANOVA) with judges, replicates, and their interactions as effects. Means were compared according to the Duncan test. Data on physical properties, proximate composition, and fatty acid profile were analysed by Student’s *t*-test for independent samples, comparing means for the two cured products (violino and bresaola). Differences were considered significant for *p* < 0.05.

A principal component analysis (PCA) was performed as an unsupervised multivariate test to detect the presence of eventual sample clustering and the relative importance of variables in determining the variability of the multivariate data matrix. Before developing the PCA, physical (L*, a*, b*, pH), chemical (ash, lipid, protein, moisture), fatty acid, and sensory data were fused, leading to a new data matrix consisting of 20 samples × 67 variables. Data were centered and scaled before performing the PCA. The scores plot, the loadings plot (PC-1 vs. PC-2), and the correlation loadings matrix were used for data interpretation. Statistical analysis was conducted with SPSS data-processing software (SPSS/PC Statistics 28.0, IBM, Armonk, New York, NY, USA). PCA was performed using Unscrambler® X version 10.4 (Camo, Oslo, Norway).

## 3. Results

### 3.1. Study Area and Animals

The flock was tracked by GPS and consisted of 1300 sheep and 200 lambs of the Bergamasca breed, with a total grazing area of 736 ha. The average grazing altitude for the 69 grazing days was 1838 m. Figure 1 shows some data obtained from the flock geolocation system for the grazing periods considered. The average age of the Bergamasca sheep considered in this study was 5.2 years (range 4.8–5.9 years). The average live weight at slaughter was 94.5 kg (range 86–100 kg) with an average hot carcass weight of 44.9 and a carcass yield of 48.9%.

### 3.2. Products’ Physical Characteristics

The average weight of bresaola and deboned violino was 427 g and 840 g, respectively, with a seasoning loss of 48% and 42%. The physical parameters, pH values, and colour indexes of the two dry-salted products are shown in Table 1.

No differences (*p* > 0.05) were found for pH and colour indexes L* and a* in the two products, violino and bresaola. The b* values were higher (*p* = 0.001) in sheep violino than in bresaola.

### 3.3. Proximate Composition

The moisture, protein, lipid, and ash content of sheep violino and bresaola are shown in Table 2. A significant difference (*p* < 0.001) was found between the two products for the moisture and ash content, which were higher in bresaola than in violino. On the other hand, the fat content was higher (*p* < 0.001) in violino than in bresaola. No difference (*p* > 0.05) was found for protein content.

### 3.4. Fatty Acid Profile

The fatty acid composition (Table 3) of Bergamasca sheep violino and bresaola provides a valuable insight into the nutritional profile of these products. Both products have a balanced composition of SFAs, monounsaturated fatty acids (MUFAs), and PUFAs, which contribute to a wide range of nutritional elements. In bresaola, SFA make a significant contribution, representing 39% of the total fatty acid content. In particular, palmitic acid (16:0) and stearic acid (18:0) are the main SFAs, accounting for 23% and 14%, respectively. The main MUFAs is oleic acid (c9-18:1), which represents 41% of the total fatty acids. In addition, PUFAs, including essential n-3 and n-6 fatty acids, contribute 10% to the total fatty acid content, with a balanced n-6/n-3 ratio of approximately 2.2.

On the other hand, deboned violino has a distinct fatty acid profile, with SFAs accounting for 43% of the total fatty acid content. Palmitic and stearic acids are the most abundant fatty acid, representing 23% and 18%, respectively. Oleic acid is the principal MUFAs, accounting for 39% of total fatty acids. PUFAs, including the essential n-3 and n-6 fatty acids, contribute 6.7% to the total fatty acids, maintaining a balanced n-6/n-3 ratio of around 2.2.

It is also important to consider the fatty acid content in the two different products. A 100 g portion of violino contains 6.6 g of SFAs and 7.0 g of MUFAs, while bresaola contains 2.7 g of SFAs and 3.4 g of MUFAs. This difference is due to the higher lipid content in deboned violino than bresaola (18.7% vs. 9.7%, respectively).

When comparing the fatty acid composition of bresaola and violino, bresaola generally has lower percentages of SFAs (38.9% vs. 48.2%) (*p* < 0.001), higher percentages of PUFAs (9.9% vs. 6.7%) (*p* < 0.001), and lower percentages of odd- and branched-chain fatty acids (OBCFAs,3.3% vs. 4.0%), while no significant differences were found for the amount of total MUFAs (47.9% vs. 46.1%) (*p* = 0.053) nor for the n-6/n-3 ratio (*p* = 0.722).

### 3.5. Sensory Profile

The mean sensory scores for appearance, aroma, taste, flavour, and textural attributes of sheep deboned violino are shown in Table 4. The *p*-values for the replicates revealed no significant differences (*p* > 0.05), and the judges showed differences in some appearance and texture descriptors (*p* < 0.05). This effect is usual in sensory evaluation and is related to the judges’ different use of the scale. Therefore, these results reveal that the mean scores for each descriptor given by the judges are suitable to evaluate the sensory profile of the sheep violino.

The mean sensory scores for appearance, aroma, taste, flavour, and textural attributes of bresaola are shown in Table 5. Also for bresaola, the *p*-values for the replicates showed no significant differences (*p* > 0.05), and the judges showed differences in some aroma and flavour descriptors (*p* < 0.05). These results, as previously observed for violino, showed that the mean score for each descriptor is appropriate for the sensory profile evaluations of the sheep bresaola.

The spider plots of the sensory profile of sheep deboned violino and bresaola are presented in Figure 2.

### 3.6. Principal Component Analysis

With the aim of exploring the importance of the variables measured, the principal component analysis was carried out on the parameters measured in the products. The first two PCs explained 53% and 10% of the total variance, respectively.

In Figure 3, the score plot shows the distribution of the samples on the plane described by the first two PCs, accounting for 63% of the total variation. The good grouping of the different products is evident along the first principal component.

## 4. Discussion

This study examined processed products made from meat from the Bergamasca sheep breed, in particular deboned violino and bresaola. The presented data showed that the use of GPS systems to monitor *grass-fed* sheep has potential for the control of flock movements during mountain pasture grazing. While the study gives some initial insights, further investigations are required to develop a more reliable link between GPS monitoring and its practical impact on the traceability of sheep products, for example, if GPS flock tracking is used as a market differentiator for violino and bresaola.

There are few studies in the literature on the quality of sheep products that reach high weights, with more emphasis on lamb or mutton products. The nutritional and sensory qualities of these products are influenced by several factors, including animal age, diet, breed, and processing methods [13,16,17,25]. In fact, the meat of adult and heavier animals has more intramuscular fat and a darker, redder colour than that of young animals and a more intense aroma typical of sheep meat [16,26].

In the present study on Bergamasca sheep, the animals were slaughtered at an average age of 5 years and an average live weight of 94 kg and there are few data about carcass characteristics of animals of this age and weight in literature [27,28]. However, this breed can reach weights significantly higher than most meat sheep and this makes it particularly interesting to produce meat and meat products.

### 4.1. Physical and Proximate Composition of Violino and Bresaola from Bergamasca Sheep

One of the important attributes in processed meat products is the weight loss during ripening. A recent study by Abi-solloum et al. [29] showed that processed sheep meat products have a higher yield than beef products. In our study, there was 42–48% seasoning loss, which is lower than the data reported in this study.

The pH values of Bergamasca sheep bresaola and violino found in our study were comparable with data previously reported in the literature for similar dry-cured sheep products [29,30]. Similarly, the colour parameters were in line with the data reported in the literature [16,29]. However, in the present study, a lower lightness value (L*) was observed in bresaola from Bergamasca sheep than in bresaola from Awassi sheep and a higher red index (a*) was found if compared to the one reported in cured sheep leg by Texteira et al. [29,31]. This aspect is mainly related to the presence in muscle tissue of myoglobin, the content of which varies by species, sex, breed, and age [32].

The proximate composition of violino and bresaola from *Bergamasca* sheep showed an average protein content of about 40%, with no differences between the two products. On the contrary, we found differences between violino and bresaola for the moisture, fat, and ash content, with violino showing the highest fat content and the lowest proportion of moisture and ash. Such differences could be attributed to the different muscles and processing conditions of the two dry cured products, as previously reported.

Generally, our results are consistent with those reported by Zhang et al. [25] for mutton and lamb bresaola, while a higher fat content (9.93% vs. 7.73%) was found in bresaola derived from Bergamasca sheep as compared to the data reported on Awassi sheep by Abi-Salloum et al. [29]. The difference is probably due to the breed used and slaughter age and weight [32]. Similarly, data on the proximate composition of Bergamasca deboned violino showed many differences from data reported in the literature. Our data showed a lower moisture content and higher protein and fat content compared to data reported by Žugić-Petrović et al. [33] in dry cured sheep hams derived from sheep raised in the Balkans and slaughtered at a live weight between 47 and 60 kg. The same differences were observed with the data reported by Texteira et al. [31] in cured sheep leg derived from the Churra Galega Bragançana breed, with a carcass weight of 20 kg. Also in this case, the differences could be attributed to the different breed and slaughter weight. In addition, different cuts and meat processing should be also considered.

### 4.2. Fatty Acid Profile Bresaola and Deboned Violino from Bergamasca Grass-Fed Sheep

The results of our study are in line with previous studies investigating the effect of pasture on lamb meat quality [34,35]. Bresaola and violino, both derived from *grass-fed* animals, showed the characteristic fatty acid composition seen in meat from *grass-fed* animals.

Feeding ruminants fresh herbage leads to an increase in branched chain fatty acids (BCFAs), linolenic acid, vaccenic acid, and rumenic acid. Particularly, sheep exhibit a large range of forage plant selection during grazing and the high botanical diversity that characterises herbage species in mountain grasslands could result in the selection of plants with elevated nutritional value, enriched in PUFAs [36]. In the rumen, PUFAs undergo biohydrogenation and saturation, facilitated by rumen bacteria, leading to the formation of SFAs. Throughout this process, numerous intermediate FAs are formed, including conjugated isomers of linoleic acid (CLAs), with rumenic acid comprising about 80%, and *trans* isomers of C18:1, such as vaccenic acid (*trans*11-18:1). A portion of these fatty acids, considered as markers for ruminal metabolism, are subsequently absorbed by the sheep’s body, assimilated into its own metabolism, and incorporated into fat deposition, including intramuscular fat. Lamb is considered the richest meat source of CLAs and *grass-fed* animals exhibit up to 1.5-fold higher CLAs concentrations compared with that from grain-fed animals [37]. Similarly, OBCFAs originate from the rumen environment as significant components of ruminal bacteria, ultimately finding their way into ruminant products—both dairy and meat—via absorption into sheep metabolism and integration into intramuscular (and milk) fats. Notably, BCFAs are highly prevalent in the membranes of cellulolytic bacteria, and products derived from *grass-fed* animals exhibit significantly higher amounts [38].

Interestingly, when found in high amounts, other than being markers of fresh herbage feeding systems, the above-mentioned fatty acids are associated with nutritional outcomes. Actually, BCFAs and rumenic acid have been demonstrated to have positive effects on human health, including cardiovascular and gut health, a better regulation of the general inflammatory status of the body, and cell functions. At the same time, linolenic acid, as the main representative of the n-3 fatty acids series and precursor of the long chain n-3 fatty acids (eicosapentaenoic acid—EPA, docosapentaenoic acid—DPA, docosahexaenoic acid—DHA), may be related to anti-atherogenic properties, with a positive effect on cholesterol level and improvement of immune response [39]. From a nutritional standpoint, bresaola and violino are noteworthy for their heightened lipid content. Fatty acid analysis revealed a substantial presence of n-3 fatty acids and an appropriate n-6/n-3 ratio, consistent with nutritional guidelines. Particularly, the content of rumenic acid is noteworthy, averaging around 100–200 mg per 100 g serving.

Our findings regarding the fatty acids composition of violino and bresaola from Bergamasca sheep align with those reported by Teixeira et al. [31] for cured legs from Churra Galega Bragançana adult sheep (8–12 years old) raised in an extensive system. Teixeira et al. [31] documented a fatty acids profile comprising 45%SFAs (~2.7 g/100 g of product), 47%MUFAs (~2.8 g/100 g of product), and 8% PUFAs (~471 mg/100 g of product). At the same time, our results appear more favourable than those recently presented by Zioud et al. [40], who investigated the impact of various processing methods on the drying of traditional products made from dry-cured lamb meat (approximately 7 months old). Regardless of the drying method employed, the authors noted 0.3% linoleic acid, 5.2–5.6% linoleic acid, and an n-6/n-3 ratio ranging from 10.1 to 11.6 in the dried product. Notably, our study revealed a lower n-6/n-3 ratio, standing at 2.2 for both violino and bresaola, even lower than the ratio of 3.94 reported by Teixeira et al. [31]. The n-6/n-3 ratio is commonly used by food and nutritional organisations to assess the nutritional value of fat in the human diet and to provide dietary recommendations. It is suggested that the n-6/n-3 ratio should not exceed the value of 4:1 [41]. According to these guidelines, the ratio reported in our study for violino and bresaola from *grass-fed* Bergamasca sheep was fully below the suitable limit. Other authors previously observed an increase in the content of n-3 fatty acids and consequent decrease in n-6/n-3 ratio in grazing lamb meat, suggesting pasture rearing as a technique for improvement of meat dietetic quality [42].

Considering the two different products analyzed in this study, the substantial difference in fat content significantly contributes to variations in their primary fatty acid composition. Interestingly, our results diverge from those of Fowler et al. [43] and Angood et al. [44] who found that increased total fat in lamb cuts correlated with higher concentrations of MUFAs and SFAs, with a lesser impact on PUFAs. In contrast, our study reveals a distinct pattern: although bresaola has lower fat content, it exhibits lower SFAs and higher MUFAs compared to violino. Additionally, bresaola has a higher content of PUFAs than violino. The observed variations in fatty acids between products may be linked to differences in the intramuscular and intermuscular fat composition in the cuts used for their preparation. However, the lack of fatty acid analysis on fresh cuts used for product preparation makes it challenging to confirm this as the cause of the observed differences.

### 4.3. Sensory Parameters of Bergamasca Sheep Products

Sensory qualities including appearance, aroma, flavour, and texture play a crucial role in the consumer’s choice and product consumption [45]. Regarding appearance, the red colour of muscle, the white colour of fat, and the homogeneity of the slice are the most important descriptors for consumers. High values were observed in both bresaola and violino for the red colour of the muscle. Conversely, the white colour of the fat and the homogeneity of the slice had average and similar values in both products, in agreement with the data reported by Texteira et al. [31] and Stojkovic et al. [26] for sheep violino. 

Bresaola and violino are cured products with a unique aroma and flavour that are correlated with the nutritional and fatty acid composition of sheep meat, the use of spices, additives, and the production techniques [45,46]. Considering sheep products, consumers generally consider the flavour, followed by tenderness and juiciness, important [47]. 

Regarding bresaola, the sensory analysis demonstrated significant scores for the wild (7.5 and 6.4, respectively) and seasoned (7.3 and 6.4, respectively) aroma and flavour descriptors, which are characteristic of sheep meat. Violino also exhibits the highest values for flavour and aroma attributes of both wild (9.2 and 8.8, respectively) and cured (7.6 and 7.7, respectively), consistent with previous studies where sensory characterisation of sheep hams showed high intensity for the cured aroma and flavour [26,31]. These outcomes for bresaola and violino could be attributed to the presence of volatile compounds that originate from lipids, nucleotides, peptides, and amino acid precursors [45,48]. These compounds can positively influence the aroma and characteristic flavour of processed products [49]. It is also observed that meat from *grass-fed* animals showed a better flavour than meat from concentrate-fed animals due to a lower presence of adverse volatile compounds [50].

Both Bresaola and violino have a high value for the tenderness descriptor (6.8 and 5.7, respectively). It is reported that the tenderness of the product is an important sensory characteristic for consumers. It is related to several factors such as age, sex and feeding of the animal, meat quality, and production technologies [50,51,52].

Sensory characteristics such as aroma and flavour are more pronounced in mature animals and may be less pleasing to certain consumers than those of younger animals [52]. Consequently, it is likely that lamb meat, which is derived from younger animals, has a greater market presence. However, processed products made from heavier and older animals have been found to enhance the commercial value of sheep meat and encourage its consumption [53].

### 4.4. Multivariate Analysis

Multivariate analysis performed through PCA revealed that a combination of the first two components (PC-1 and PC-2) explained 63% of variability in the data matrix, with PC-1 accounting for 53% of variance and PC-2 for 10%. Actually, violino and bresaola samples partially separated along the direction of PC-1, while PC-2 had no effect on any clustering or sample discrimination. The analysis of the correlation matrix for loadings revealed that the variables associated to the highest weight in PC-1 were lipid content of samples and fatty acids amounts, which were mostly distributed in the negative quadrants (−) for the first component; on the contrary, water content and two sensory parameters (s8, s17) were distributed in the positive quadrants (+) for the first component.

In the scores plot, violino samples were distributed in the area of the bi-dimensional space associated with negative values for PC-1, in positive correlation with the distribution of most fatty acids and lipid content. On the contrary, bresaola samples were distributed in the area of the plot associated with positive values for PC-2, in positive correlation with moisture and ash content, arachidonic acid (fa28), softness (s17), and spicy aroma (s8).

Finally, some correlations were detected among fatty acids content and sensory descriptors. In detail, 14:0, *iso-*14, *anteiso-*15, 15:0, *iso-*16, 17:0, and *trans*11-18:1 showed correlation indexes higher than 0.70 with the *wild aroma* descriptor, with *iso-*14 and *anteiso-*15 showing a positive correlation (correlation index = 0.72) also with the *wild flavour* descriptor. Further, *iso-*14 showed a positive correlation (correlation index = 0.76) with the *global intensity aroma* descriptor. These outcomes are very interesting since they suggested that higher amounts of specific fatty acids, mainly identified as the BCFAs group, may be related to higher perception of *wild aroma* and *flavour* in dried sheep products from *grass-fed* Bergamasca sheep. This matches what was previously reported regarding the contribution of BCFAs towards the distinct mutton-like aroma in lamb fat and muscle tissues, due to the presence of this unique class of fatty acids, which derive from both ruminal fermentations and triacylglycerol hydrolysis occurring in lamb fat during product processing and seasoning [54]. 

## 5. Conclusions

The nutritional characterisation of dry-salted products, bresaola and violino, from Bergamasca sheep meat, showed their positive attributes for human consumption. In fact, both products have a high protein content and a favourable fatty acid profile, with a healthy PUFA/SFA ratio and n-6/n-3 ratio. The sensory evaluation showed that the aroma and flavour of the two seasoned products were typical of the ovine species. The characteristics of seasoned products derived from the Bergamasca sheep can be valorised with a traceable, green, and sustainable supply chain, considering both the monitoring of the pasture with GPS and the *grass-fed* animals. The presented data suggest that utilising GPS systems for monitoring *grass-fed* sheep offers the potential for real-time control of flock movements during grazing in mountain pastures. This research has the potential to yield valuable information for consumers, but additional empirical evidence and exploration are needed to fully support this claim. The study emphasises the potential for added value to seasoned products from Bergamasca sheep through a traceable, sustainable, and health-conscious supply chain.

## Figures and Tables

**Figure 1 animals-14-00488-f001:**
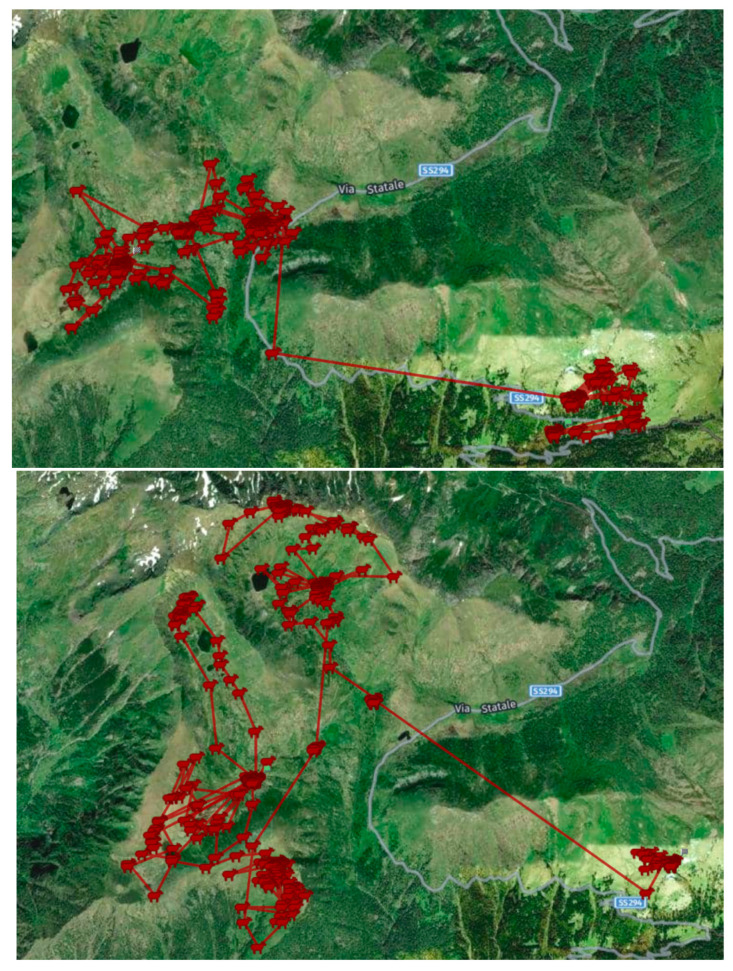
Flock route in July and August 2021 in Scalve Valley (Bergamo, Italy).

**Figure 2 animals-14-00488-f002:**
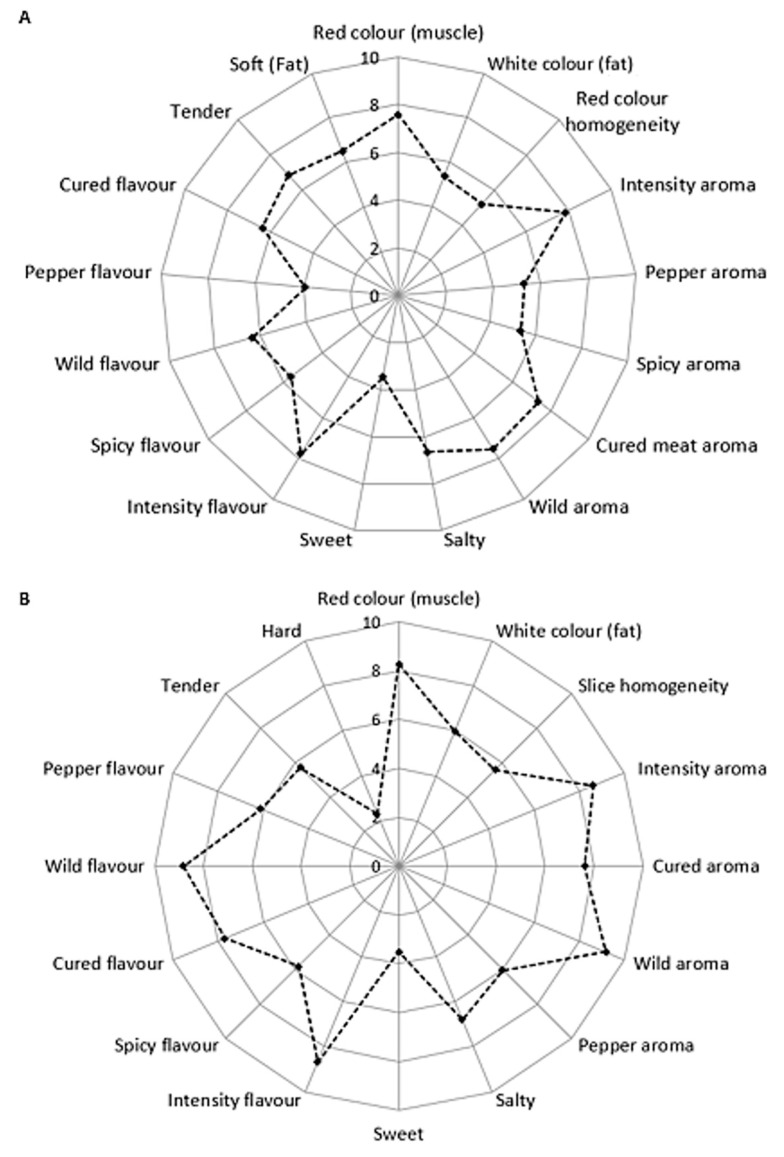
Spider plots of the sensory profile of sheep violino (**A**) and bresaola (**B**).

**Figure 3 animals-14-00488-f003:**
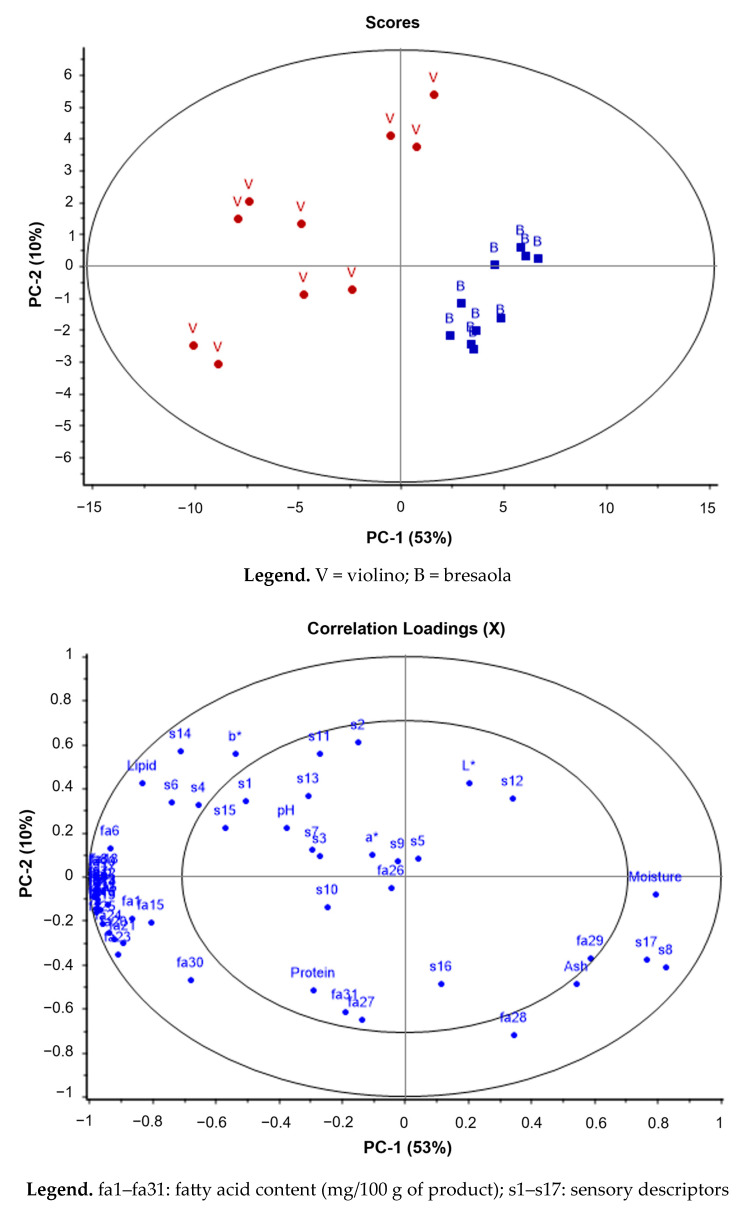
PCA score plot (**top**) and loadings plot (**bottom**).

**Table 1 animals-14-00488-t001:** Physical parameters of the Bergamasca sheep deboned violino and bresaola (*n* = 10).

Item ^1^	Violino	Bresaola	SEM	*p*-Value
pH	6.01	5.73	0.1	0.058
Colour parameters:				
L*	27.6	28.3	1.5	0.827
a*	10.2	10.4	1.4	0.897
b*	6.0	4.2	0.4	0.001

^1^ Data are reported as mean and standard error of the means (SEM).

**Table 2 animals-14-00488-t002:** Proximate composition (on wet weight) of the Bergamasca sheep deboned violino and bresaola (*n* = 10).

Item ^1^	Violino	Bresaola	SEM	*p*-Value
Moisture, %	33.25	42.39	1.2	<0.001
Protein, %	40.65	40.73	0.9	0.915
Lipid, %	18.73	9.75	0.4	<0.001
Ash, %	5.99	6.14	0.3	<0.001

^1^ Data are reported as mean and standard error of the means (SEM).

**Table 3 animals-14-00488-t003:** Fatty acid profile of the Bergamasca sheep deboned violino and bresaola. Individual FAs and their sum (ΣSFAs, ΣMUFAs, ΣPUFAs, *iso*-FAs, *anteiso*-FAs, odd-chain FAs, ΣOBCFAs) are expressed as concentration (mg/100 g) and % by weight of total fatty acids (*n* = 10). Data are reported as mean, SEM, and *p*-value for the *t*-statistic.

	Violino		Bresaola			
Fatty Acid	mg/100 g	% of Total FAs	mg/100 g	% of Total FAs	SEM *t*-Test	*p*-Value *t*-Test
12:0	12.4	0.08	6.1	0.09	1.7	0.002
14:0	328.8	2.17	130.3	1.83	35.9	<0.001
16:0	3432.7	22.61	1635.5	23.01	364.2	<0.001
18:0	2852.6	18.23	985.9	13.93	384.1	0.001
20:0	17.1	0.11	5.6	0.08	2.8	0.002
ΣSFAs	6643.5	43.2	2763.4	38.9	767.3	0.001
16:1	212.6	1.44	132.4	1.83	25.2	0.005
17:1	100.3	0.66	50.6	0.70	10.7	0.001
t9, 18:1	88.2	0.57	32.9	0.46	10.9	<0.001
t11, 18:1	553.5	3.53	162.8	2.36	78.6	<0.001
c9, 18:1	5904.6	38.89	2935.9	41.15	635.1	0.001
c11, 18:1	141.2	0.94	91.6	1.29	14.2	0.004
20:1	11.3	0.07	6.6	0.09	1.5	0.011
ΣMUFAs	7011.8	46.1	3412.6	47.9	753.7	0.001
t9t12, 18:2n-6	87.0	0.55	28.3	0.39	11.9	<0.001
c9c12, 18:2n-6	410.1	2.78	310.3	4.44	34.4	0.012
c9t11, 18:2 (CLA)	184.9	1.20	92.8	1.29	23.2	0.002
20:2n-6	2.0	0.01	1.9	0.03	0.5	0.904
18:3n-3	174.6	1.16	111.1	1.57	18.8	0.006
20:3n-6	5.8	0.04	6.5	0.10	0.6	0.266
20:4n-6	55.3	0.40	66.9	0.97	3.9	0.007
20:5n-3 (EPA)	19.6	0.14	25.0	0.37	1.5	0.002
22:5n-3	48.8	0.33	39.0	0.54	6.4	0.145
22:6n-3 (DHA)	10.2	0.07	10.9	0.15	2.1	0.726
ΣPUFAs	998.2	6.7	692.7	9.9	95.2	0.007
n-6	560.1	1.29	414.0	5.9	48.0	0.010
n-3	253.2	0.84	185.9	2.6	25.4	0.019
n-6/n-3	2.2	2.2	2.2	2.2	0.1	0.725
P/S	0.09	0.09	0.15	0.15	0.0	<0.001
*iso*-14:0	9.3	0.06	1.6	0.02	1.3	<0.001
*iso*-15:0	39.3	0.26	16.5	0.23	4.3	<0.001
*anteiso*-15:0	27.5	0.18	10.1	0.14	3.4	<0.001
n-15:0	76.2	0.49	28.9	0.41	9.6	0.001
*iso*-16:0	37.5	0.24	14.6	0.20	4.6	<0.001
*iso*-17:0	65.2	0.42	29.0	0.41	7.9	0.001
*anteiso*-17:0	103.0	0.67	38.7	0.54	12.8	0.001
n-17:0	217.0	1.39	75.9	1.06	28.6	0.001
*iso*-18:0	46.2	0.30	21.8	0.31	5.0	0.001
*iso*-FAs	197.5	1.29	83.5	1.17	22.1	<0.001
*anteiso*-FAs	130.5	0.84	48.9	0.68	16.0	<0.001
odd-chain FAs	293.2	1.88	104.8	1.47	38.0	0.001
ΣOΒCFAs	621.2	4.0	237.1	3.3	75.7	0.001

SFAs, saturated fatty acids; MUFAs, monounsaturated fatty acids; CLA, conjugated linoleic acid; PUFAs, polyunsaturated fatty acids; OBCFAs, odd- and branched-chain fatty acids; P/S = 18/2n-6 + 18:3n-3/14:0 + 16:0 + 18:0.

**Table 4 animals-14-00488-t004:** Sensory evaluation of sheep violino and statistical significance of judges (*n* = 10) and replicates (*n* = 3) for each sensory descriptor.

Descriptors	Mean	SEM	*p-*Value Judges	*p-*Value Replicates
Appearance				
Red colour (muscle)	8.3	0.204	0.035	0.618
White colour (fat)	6.0	0.950	0.001	0.449
Slice homogeneity	5.6	0.440	0.001	0.950
Aroma				
Global intensity	8.6	0.169	0.152	0.202
Cured	7.6	0.267	0.076	0.063
Wild	9.2	0.114	0.066	0.156
Pepper	6.0	0.516	0.007	0.672
Taste				
Salty	6.8	0.360	0.084	0.954
Sweet	3.5	0.386	0.001	0.847
Flavour				
Global intensity	8.7	0.172	0.001	0.620
Spicy	5.8	0.633	0.197	0.120
Cured	7.7	0.267	0.179	0.744
Wild	8.8	0.302	0.098	0.782
Pepper	6.1	0.395	0.074	0.214
Texture				
Tenderness	5.7	0.366	0.001	0.522
Hardness	2.3	0.323	0.001	0.715

**Table 5 animals-14-00488-t005:** Sensory evaluation of sheep bresaola and statistical significance of judges (*n* = 10) and replicates (*n* = 3) for each sensory descriptor.

Descriptors	Mean	SEM	*p-*Value Judges	*p-*Value Replicates
Appearance				
Red colour (muscle)	7.6	0.203	0.075	0.476
White colour (fat)	5.4	0.539	0.931	0.051
Red colour homogeneity	5.1	0.285	0.127	0.492
Aroma				
Global intensity	7.8	0.256	0.115	0.439
Pepper	5.3	0.351	0.001	0.727
Spicy	5.3	0.471	0.001	0.143
Cured meat	7.3	0.295	0.051	0.073
Wild	7.5	0.321	0.178	0.545
Taste				
Salty	6.6	0.276	0.014	0.199
Sweet	3.4	0.340	0.694	0.127
Flavour				
Global intensity	7.8	0.233	0.010	0.468
Spicy	5.6	0.293	0.250	0.811
Wild	6.3	0.331	0.003	0.111
Pepper	3.9	0.379	0.001	0.444
Cured	6.4	0.306	0.004	0.905
Texture				
Tenderness	6.8	0.312	0.120	0.201
Softness (fat)	6.5	0.206	0.312	0.831

## Data Availability

Research data will be provided upon request.

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
