# Peer review of "Characterisation of Dry-Salted Violino and Bresaola from Grass-Fed Bergamasca Sheep"

_animals, 2024, doi:10.3390/ani14030488_

Round 1

Reviewer 1 Report

Comments and Suggestions for Authors

Title: Characterization of two dry-cured products from grass fed Bergamasca sheep tracked with GPS system (Animals – ID 2818647) – Reviewer #.

English: The manuscript needs to be revised as there are many mistakes, also in terms of punctuation and style.

Title – The inclusion of “GPS system” in the title seems not to be adequate considering the main topics and results of the manuscript (see also specific comments)

Simple summary – Add more details about the main topics of the study and revise according to the specific comments; lines 21-23 seems out of the goal of the study

Abstract - Revise according the specific comments and more results

Introduction - Revise according the specific comments

Results and discussion – Results are sometimes not clear. Discussion must be improved.

Conclusions Improve taking into account the previous comments – The role of GIS is poor exploited in the manuscript while there is a great emphasis on the conclusions.

References – Some references (n° 7, 8, 9, 12, 35, 38) are dated or even too much dated (n° 10, 15) ... may be better to refresh these with recently ones

Reference n° 7 is the same of n° 26; reference n° 10 is the same of n° 28.

SPECIFIC COMMENTS

Line 2. No full stop at the end of title

Line 47 May be better “approximately 6 million“ instead of 6,122,117  

Lines 74-86. May be better move this period at line 62 to complete the part related to “sheep”

Line 88-90. Most of the information is not relevant to the study’s objective (especially L. 45-73). I also think the authors do not explain and justify: i. why they use GPS (why is it so important/what is the benefit of using GPS for the objectives of the study?); ii. Why are they comparing nutritional and sensory characteristics? Iii. What is the connection among the nutritional and sensory characteristics that are investigated, the grass-fed system and the traceable/sustainable supply chain that the consumer is driven towards? Therefore, the objectives should be better describe.

Line 112. ‘160 x 5 cm’

Line 121. Which company about SPOT LLC?

Line 129 °C

Line 131 ‘n = 10’

Line 141. 45% (not 40-50 %)… also the symbol % attached to the number

Line 142 not C° but °C …. And also 12 °C (leave space between number and °C) and so on…

Line 142. 90 months ?! you mean 9 months I guess

Lines 150-151. Not necessary ‘The …. Milan.’

Line 172. AOAC is ‘Association of Official Analytical Chemists (AOAC)’

Line244-245. DMI of the flock …not clear (see next comment about Table 1.)

Line 245. Some details about the pasture? Botanical composition, chemical composition (crude protein, NDF, starch)?

Table 1. What is ‘Flock kg DM/d’? It’s a daily dry matter intake (of what?). If DMI, how did you perform this data?

Line 258 (Figure 1). Which year?

Table 2. Move ‘n = 10’ in the title. SEM or SD (standard deviation)? If SEM, from which statistical model?

Lines 261-263. Write results in text or in table 2 (not both)

Table 2. Avoid too much decimals (enough just on decimal for colour parameters)

Table 3. add n = 10 in the title; ‘Proximate composition (on wet weight) of ….’ . Therefore delete the footnote.

Line 305. (p < 0.001) …and so on …. (don’t use P as capital letter)

Line 311 and 317. ‘(mg/ 100 g)’

Table 4. Also ‘(mg/100 g)’

Table 4. Avoid the use of too much decimals: use 1 decimal until 100 g; no decimal for FA with a content higher than 100.

Line 338, 339, 340, 471. ‘100 g’, ‘6.6 g’ and so on…

Line 362. ‘p-values’

Tables 6. and 7. Perhaps just one decimal for the mean values?

L. 388-393. Here the authors talk about the use of GPS drawings conclusions for which I do not see any previous connections/explanation/justification with/in the rest of the manuscript.

L. 395-405. Redundant information that often would be more suitable for the Introduction section.

Lines 452-454 Too much generic. Improve

Lines 456-466 It’s redundant and it’s too much speculative, re-write and link to your research.

Line 472. The authors should add more about the impact that the diet (more information needs to be provided in the methods) has on the FA profile

Lines 497-499. It’s sound introduction. Re-write

L. 509-512.  Conclusion not justified in results&discussion

Comments on the Quality of English Language

English: The manuscript needs to be revised as there are many mistakes, also in terms of punctuation and style.

Author Response

Animals 2818647R1

Title: Characterization of dry-salted violino and bresaola from grass fed Bergamasca sheep.

Response to reviewer 1.

The authors express their sincere gratitude to the reviewer for providing invaluable insights and constructive comments, which have significantly enhanced the quality of the paper. All the suggested change has been reported in red font and has been reported below:

  1. English: The manuscript needs to be revised as there are many mistakes, also in terms of punctuation and style.

The English has undergone revision for clarity and correctness.

  1. Title – The inclusion of “GPS system” in the title seems not to be adequate considering the main topics and results of the manuscript (see also specific comments)

The title has been revised, as also suggested by other two reviewers.

  1. Simple summary – Add more details about the main topics of the study and revise according to the specific comments; lines 21-23 seems out of the goal of the study

The simple summary has been revised.

  1. Abstract - Revise according the specific comments and more results

The abstract has been revised.

  1. Introduction - Revise according the specific comments

The introduction has been revised as suggested.

  1. Results and discussion – Results are sometimes not clear. Discussion must be improved.

Results section has been revised, also in in accordance with the feedback provided by reviewers 2 and 3. The Discussion section has been improved.

  1. Conclusions Improve taking into account the previous comments – The role of GIS is poor exploited in the manuscript while there is a great emphasis on the conclusions.

The role of GPS in the conclusions has been downsized, and the overall conclusion section has been improved, incorporating valuable insights from the feedback received.

  1. References – Some references (n° 7, 8, 9, 12, 35, 38) are dated or even too much dated (n° 10, 15) ... may be better to refresh these with recently ones.

Some references have been delete from the text and was replaced with recently ones.

  1. Reference n° 7 is the same of n° 26; reference n° 10 is the same of n° 28.

All the reference section has been revised.

SPECIFIC COMMENTS

All the suggested change has been made and reported below:

Line 2. No full stop at the end of title

New Line 3: Full stop has been added.

Line 47 May be better “approximately 6 million“ instead of 6,122,117  

New Line 48: The suggested change has been made.

Lines 74-86. May be better move this period at line 62 to complete the part related to “sheep”

New Line 64-69: The suggested change has been made.

Line 88-90. Most of the information is not relevant to the study’s objective (especially L. 45-73).

New Line 92-100: To address this concern, we have carefully revisited and revise this section, ensuring that the content aligns more closely with the study's objectives. Our aim is to provide a clearer and more concise presentation of information that directly contributes to the overall understanding of our study.

I also think the authors do not explain and justify:

  1. why they use GPS (why is it so important/what is the benefit of using GPS for the objectives of the study?)

Integrating GPS systems into sheep farming operations enhances the overall quality, welfare, and traceability of sheep and derived products. It provides farmers with valuable data for informed decision making, improves efficiency in flock management, and meets the growing demand for transparency in the food supply chain. In our study, GPS flock tracking can be used as a market differentiator. Consumers increasingly value transparency and sustainable practices in meat production, and GPS technology contributes to these goals. This topic has been added to the introduction section (New lines 95-97).

  1. Why are they comparing nutritional and sensory characteristics?

The two products derived from different meat cuts and were prepared using different techniques and ripening times. For these reasons, they were compared with each other. The objectives of the study have been refined in the text.

  1. What is the connection among the nutritional and sensory characteristics that are investigated, the grass-fed system and the traceable/sustainable supply chain that the consumer is driven towards? Therefore, the objectives should be better describe.

As mentioned above, the objectives were re-written and better described (New lines 92-100).

Line 112. ‘160 x 5 cm’

New line 120: The suggested change has been made.

Line 121. Which company about SPOT LLC?

New line 129: The company has been added.

Line 129 °C

New line 136: The suggested change has been made.

Line 131 ‘= 10’

NL 138: The suggested change has been made.

Line 141. 45% (not 40-50 %)… also the symbol % attached to the number

New lines 144-165: The paragraph has been revised, according also the reviewer 2 comments and the suggested change has been made.

Line 142 not C° but °C …. And also 12 °C (leave space between number and °C) and so on…

New lines 144-165: The paragraph has been revised, according also the reviewer 2 comments and the suggested change has been made.

Line 142. 90 months ?! you mean 9 months I guess

New line 165: The authors apologize for the mistake, the correct length is 3 months (90 days).

Lines 150-151. Not necessary ‘The …. Milan.’

The sentence has been deleted.

Line 172. AOAC is ‘Association of Official Analytical Chemists (AOAC)’

New line 188: The suggested change has been made.

Line244-245. DMI of the flock …not clear (see next comment about Table 1.)

All the part about movement of the sheep tracked using GPS are now reported in the text and some data has been deleted according to the comments of Reviewer 2.

Line 245. Some details about the pasture? Botanical composition, chemical composition (crude protein, NDF, starch)?

Unfortunately, a vegetation survey for botanical data collection and their chemical composition was not performed in this study.

Table 1. What is ‘Flock kg DM/d’? It’s a daily dry matter intake (of what?). If DMI, how did you perform this data?

This data has been deleted from the text, according to the comments of Reviewer 2.

Line 258 (Figure 1). Which year?

Figure 1. The years has been inserted.

Table 2. Move ‘n = 10’ in the title. SEM or SD (standard deviation)? If SEM, from which statistical model?

Table 2 has been deleted and the mean data with the range have been now reported in the text (New Lines 263-269) according with the comments of reviewer 2.

Lines 261-263. Write results in text or in table 2 (not both)

Table 2 has been deleted and the mean data with the range have been now reported in the text (New Lines 263-269).

Table 2. Avoid too much decimals (enough just on decimal for colour parameters)

Table 1. The suggested change has been made.

Table 3. add n = 10 in the title; ‘Proximate composition (on wet weight) of ….’ . Therefore delete the footnote.

Table 2. The suggested change has been made.

Line 305. (p < 0.001) …and so on …. (don’t use P as capital letter)

New Lines 288-290: The suggested change has been made.

Line 311 and 317. ‘(mg/ 100 g)’

The paragraph has totally revised.

Table 4. Also ‘(mg/100 g)’

New Lines 318: Table 3. The suggested change has been made.

Table 4. Avoid the use of too much decimals: use 1 decimal until 100 g; no decimal for FA with a content higher than 100

Table 3. The tables reported the FA composition of the two dry cured product has been revised according the suggestion of reviewer 2. The second decimal number has been deleted.

Line 338, 339, 340, 471. ‘100 g’, ‘6.6 g’ and so on…

New lines 327-330: The suggested change has been made.

Line 362. ‘p-values’

New lines 350:  The suggested change has been made.

Tables 6. and 7. Perhaps just one decimal for the mean values?

New Tables 4 and 5 The suggested change has been made.

  1. L. 388-393. Here the authors talk about the use of GPS drawings conclusions for which I do not see any previous connections/explanation/justification with/in the rest of the manuscript.

The sentences have been deleted from the manuscript, as also suggested by reviewer 2

  1. 395-405. Redundant information that often would be more suitable for the Introduction section.

Upon a thorough review, we acknowledge that some information in this section could be perceived as redundant and may find a more suitable placement in the introduction section. We understand the importance of streamlining the content to enhance clarity and avoid unnecessary repetition. It is also important to poin outh that in literature there are few studies regarding animals of these age/weight, as now reported in new  lines 386-391.

Lines 452-454 Too much generic. Improve

Lines 456-466 It’s redundant and it’s too much speculative, re-write and link to your research.

In our revised manuscript, we have reevaluated the content in lines 395-466 and assess its relevance in the results section. Any information that is more appropriate for the introduction has been relocated accordingly, ensuring a more cohesive and focused presentation of our findings.

Line 472. The authors should add more about the impact that the diet (more information needs to be provided in the methods) has on the FA profile.

Regrettably, in this study we did not determine the fatty acid composition of herbs and botanical essences fed by animals. However, we recognize the significance of addressing this aspect, and we have enhanced the manuscript by incorporating relevant literature data that may explain the potential impact of the diet on the FA profile (new Lines 436-459).

Lines 497-499. It’s sound introduction. Re-write

New Lines 526-529: We have reevaluated the content in lines 497-499

  1. 509-512.  Conclusion not justified in results&discussion

New Lines 569-582: We recognize that the linkage between the results presented and the concluding statements in lines 509-512 needs further clarification. In our revised manuscript, we have revisited it to ensure a more explicit and justified connection between the findings presented in the results section and the conclusions.

Reviewer 2 Report

Comments and Suggestions for Authors

Characterization of two dry-cured products from grass fed Bergamasca sheep tracked with GPS system

Review 1

General comments:

The manuscript investigates the physical chemical and sensory properties of two different dry-cured mutton (produced from the meat of older sheep categories), named violin and bresaola. For their production, the meat of grass-fed Bergamasca sheep breed tracked with GPS was used. The possibility of using GPS in tracking the movement of sheep is very interesting and useful, which the authors explained in detail in the manuscript. However, tracking the movement of sheep has no significant relationship with the results presented in the manuscript. Namely, the research is not based on the influence of the grazing location on the investigated product properties, but on the differences in the production technology of two different meat products.

However, since the research results are interesting and worth publishing, I suggest the authors reorganize the manuscript as follows:

- in the description of the method of sheep breeding, state the breeding area and mention that the movement of the sheep is tracked using GPS.

- The introduction should be based on production data, technological and other differences between violino and bresaola products. The aim of the research should be to determine the effect of technological differences on the physical chemical and sensory properties of dry mutton.

- In the M&M chapter, explain in detail the production technology of these two products. I recommend statistically processing the relationship between the profile of fatty acids and the sensory properties of the products (PCA analysis).

- in the Discussion chapter, explain the differences that are obviously the result of different technologies, since the meat of the same sheep (left and right half) was used.

- The title of the manuscript should be adapted, too

I am not an expert in the English language, but I recommended checking for typographical errors, punctuation, and especially grammar throughout the manuscript.

Some specific comments are placed directly in the attached manuscript.

Author Response

Animals 2818647R1

Title: Characterization of dry-salted violino and bresaola from grass fed Bergamasca sheep.

Response to reviewer 2.

General comments:

  1. The manuscript investigates the physical chemical and sensory properties of two different dry-cured mutton (produced from the meat of older sheep categories), named violin and bresaola. For their production, the meat of grass-fed Bergamasca sheep breed tracked with GPS was used. The possibility of using GPS in tracking the movement of sheep is very interesting and useful, which the authors explained in detail in the manuscript. However, tracking the movement of sheep has no significant relationship with the results presented in the manuscript. Namely, the research is not based on the influence of the grazing location on the investigated product properties, but on the differences in the production technology of two different meat products.

The authors thank the reviewer for his valuable comments. As suggested, the GPS data have been reduced. However, the tracking of animals during the transhumance period makes it possible to link the product to the area where the animals were fed. Such data could therefore be important information to be included on the product label. The authors have revised the aim of the study by introducing a comparison between the two products, which have different chemical and sensory characteristics linked to the production technology and the different cut of meat used.

All the suggested change has been reported in red font and has been reported below

  1. In the description of the method of sheep breeding, state the breeding area and mention that the movement of the sheep is tracked using GPS.

The suggested change has been made and the data about GPS localization have been reduced (new lines 363-265).

  1. The introduction should be based on production data, technological and other differences between violino and bresaola products. The aim of the research should be to determine the effect of technological differences on the physical chemical and sensory properties of dry mutton.

In our revised manuscript, we have enhanced the introduction to better emphasize these aspects. Additionally, we have better stated the aim of the research to determine the impact of technological differences on the physical, chemical, and sensory properties of dry-salted violin and bresaola (new lines 92-100).

  1. In the M&M chapter, explain in detail the production technology of these two products. I recommend statistically processing the relationship between the profile of fatty acids and the sensory properties of the products (PCA analysis).

In the M&M section, we have provided a detailed explanation of the production technology for both violino and bresaola (new lines 144-165). Additionally, we have analysed the relationship between the fatty acid profile, the chemical composition and the sensory properties, using PCA. We have incorporated this statistical approach in our revised manuscript (new lines 251-260).

  1. in the Discussion chapter, explain the differences that are obviously the result of different technologies, since the meat of the same sheep (left and right half) was used.

This part has been revised according to this suggestion.

  1. The title of the manuscript should be adapted, too

The title of the manuscript has been changed, as suggested.

  1. I am not an expert in the English language, but I recommended checking for typographical errors, punctuation, and especially grammar throughout the manuscript.

The English has undergone revision for clarity and correctness.

Some specific comments are placed directly in the attached manuscript.

  1. Abstract must be revised.

Abstract has been revised as suggested.

Line 152: move the sentence to the next paragraph.

New line 172: the sentence has been moved as suggested.

Line 158-167: it is not clear on which muscle the pH and colour was determined at 24 hours

The authors regret the error. The pH and colour indices were determined on the processed products at the time of sampling. This statement has been inserted in the text.

Table 2. The sample of Bergamasca sheep is too small to characterize carcass characteristics. Provide range of data of the sample used.

New line 266-269: As suggested the data about carcass characteristic has been reported in the text (mean value and range)

Tables of fatty acid profile: data are redundant.

The fatty acid data have been reported as a percentage of the total but have also been quantified to give an indication of the amount of fatty acids that are ingested by the consumer and expressed in g/100g of product. The data has been merged in one FA table (new table 3).

Line 387-393: Delete the sentences about the use of GPS that are not related to the aim of the study.

New lines 387-393: The data on GPS have been revised. In fact, the tracking of animals during the transhumance period makes it possible to link the product to the area where the animals were fed. Such data could therefore be important information to be included on the product label.

Line 402: there are literature on animals reach high weight.

New lines 397: The literature has been now introduced in the text.

Reviewer 3 Report

Comments and Suggestions for Authors

This is a very interesting study addressing the problem of characterising grass fed products but certain points need to be addressed before publication.

Title: please consider adding the names of the products in the title

 Introduction

Line 45: please add reference

Line 47: the number you provide is exact and not "around"

Please provide some information about the products and not "typically, these products...) line 68

Please provide some information on the current knowledge about the grass fed sheep meat products.

These sheep are quite old. Is it a common practice to use animals at these age? Were these animals kept for reproduction purposes?

Materials and methods

Which criteria were used for animal selection? These were male sheep and ewes.

Product processing

Please try to expain better the production procedure of the two products. It seems that products are made from different muscles,

Line 142: do you mean 90 days and not 90 months?

Physical parameters

You measured the pH of the sample 24 h post mortem. This is not product pH but carcass pH. As repoted in product processing the products were not ready before at least 60 days.

The same observation applies also for colour.

I guess that you prepated one product from each carcass.

Results and discussion

My major consideration is that you are comparing the two products. This is not useful since both products are made from the same carcasses. Differences in proximate composition are related to different processing procedures.

It is important that you link the products with the grazing are. You should use certain biomarkers i.e. fatty acids.

Certain characteristics such as colour are related to animal characteristics such as animal age.

Regarding the product characteristics it is important that you compare your data with data from feeding trials where the animals were fed on concentrates in order to support the finding and show why grass fed sheep produce different products.

Lines 388-389. This statement is not supported by your data.

Line 391 - 393 is part of the introduction where you can also report on studies where modern technology is applied.

The superiority of the products should be based on compositional differences with products produced from animals fed on concentrates and the also there should be a link with the grazing species.

Comments on the Quality of English Language

-

Author Response

Animals 2818647R1

Title: Characterization of dry-salted violino and bresaola from grass fed Bergamasca sheep.

Response to reviewer 3.

This is a very interesting study addressing the problem of characterising grass fed products but certain points need to be addressed before publication.

The authors thank the reviewer for his valuable comments. All the suggested change has been reported in red font and are reported below.

Title: please consider adding the names of the products in the title

The names of the products have been added to the title.

Introduction

Line 45: please add reference

The introduction has been revised, added some data on ovine sector in UE (New Lines 43-47)

Line 47: the number you provide is exact and not "around"

The number of sheep has been rounded to 6 million.

Please provide some information about the products and not "typically, these products...) line 68

The information provided is specifically relevant to these products. The detailed processing technology is provided in the Materials and Methods section (new lines 144-165).

Please provide some information on the current knowledge about the grass fed sheep meat products.

The information has been provided in the discussion section, as suggested also by other reviewers (New Lines 437-459).

These sheep are quite old. Is it a common practice to use animals at these age? Were these animals kept for reproduction purposes?

The main product of Bergamasca breeding is represented by the lamb, with a live weight at slaughter averaging 45 kg. The sheep used in this study are breeding stock (all females) at the end of their career.

Materials and methods

Which criteria were used for animal selection? These were male sheep and ewes.

The primary focus of Bergamasca breeding is on lambs, typically slaughtered at an average live weight of 45 kg. In this study, the sheep under consideration are part of the breeding stock, exclusively females, and are at the conclusion of their productive life, having fulfilled their role in the breeding program. This practice of processing thighs into salted products is traditional and allows the commercial valorization of older animals. The criteria used for animal selection were reported in the material and method section (new lines 133-134).

Product processing

Please try to explain better the production procedure of the two products. It seems that products are made from different muscles

The processing procedure was completely re-written (new lines 144-165).

Line 142: do you mean 90 days and not 90 months?

The authors regret the error. The correct value is 90 days.

Physical parameters

You measured the pH of the sample 24 h post mortem. This is not product pH but carcass pH. As repoted in product processing the products were not ready before at least 60 days.

The same observation applies also for colour. I guess that you prepared one product from each carcass.

The authors regret the error. The pH and colour indices were determined on the processed products at the time of sampling. This statement has been inserted in the text.

Results and discussion

My major consideration is that you are comparing the two products. This is not useful since both products are made from the same carcasses. Differences in proximate composition are related to different processing procedures.

It is important that you link the products with the grazing are. You should use certain biomarkers i.e. fatty acids.

Certain characteristics such as colour are related to animal characteristics such as animal age.

Regarding the product characteristics it is important that you compare your data with data from feeding trials where the animals were fed on concentrates in order to support the finding and show why grass fed sheep produce different products.

We recognize that both products derived from the same carcass, and any differences in proximate composition are likely to be attributed to distinct meat cuts and processing procedures. To address this concern, we have refined the processing procedures with the aim to clarify the origin of any observed differences in composition. Furthermore, we understand the significance of linking product characteristics with the grazing area. Regrettably, in this study we did not determine the fatty acid composition of herbs and botanical essences fed by animals. However, we have extended our discussion to include comparative insights from relevant feeding trials, thereby elucidating the unique attributes of sheep products derived from grass-fed sheep (New Lines 437-459).

Lines 388-389. This statement is not supported by your data.

This statement has been deleted.

Line 391 - 393 is part of the introduction where you can also report on studies where modern technology is applied.

This part has been deleted from the discussion section.

The superiority of the products should be based on compositional differences with products produced from animals fed on concentrates and the also there should be a link with the grazing species.

To address the concern about the superiority of the products, we have emphasized compositional differences, particularly in comparison to products derived from animals fed on concentrates.

Round 2

Reviewer 2 Report

Comments and Suggestions for Authors

Dear authors,

Thank you for your efforts! You did a great job and improved the article significantly! The article is now correct and ready to publish.

However, I noticed that you do not follow the rule regarding the use of abbreviations. The full name of the term with an abbreviation in the parentheses, eg Polyunsaturated Fatty Acids (PUFAs) should only be given the first time it appears in the text. Later only the abbreviation should be used, or if there is a purpose (eg better understanding) only the full name without the abbreviation, not both as eg in lines 444, 453, 456,498, etc. If the editors tolerate it, OK, but if not, then this should be corrected throughout the manuscript before publication.

Kind regards

Reviewer 3 Report

Comments and Suggestions for Authors

The manuscript is fine. Just do not use the word "career" for use (line 51). Use at the end of their productive life.